# The Mechanical and Antibacterial Properties of Boron Nitride/Silver Nanocomposite Enhanced Polymethyl Methacrylate Resin for Application in Oral Denture Bases

**DOI:** 10.3390/biomimetics7030138

**Published:** 2022-09-19

**Authors:** Miao Li, Sifan Wang, Ruizhi Li, Yuting Wang, Xinyue Fan, Wanru Gong, Yu Ma

**Affiliations:** 1School/Hospital of Stomatology, Lanzhou University, Lanzhou 730000, China; 2Key Laboratory of Dental Maxillofacial Reconstruction and Biological Intelligence Manufacturing, Lanzhou University, Lanzhou 730000, China

**Keywords:** denture base, boron nitride, silver, nanoparticles, mechanical properties, antibacterial, cytotoxicity

## Abstract

The introduction of nanomaterials into polymethyl methacrylate (PMMA) resin has been effective for mechanically reinforcing PMMA for application in oral denture bases. However, these methods cannot simultaneously improve the mechanical and antibacterial properties, which limits widespread clinical application. Here, we self-assembled binary nanocomposites of boron nitride nanosheets (h-BNNs) and silver nanoparticles (AgNPs) as nanofillers and incorporated the nanofillers into PMMA. The aim of this study was to achieve antibacterial effects while significantly improving the mechanical properties of PMMA and provide a theoretical basis for further clinical application. We employed scanning electron microscopy (SEM), X-ray diffraction (XRD), Raman spectroscopy (Raman), Ultraviolet visible spectrum (UV) and atomic force microscopy (AFM) to investigate the microscopic morphology and composition of PMMA containing nanocomposites with different mass fraction. In addition, the content of the h-BNNs/AgNPs was 1 wt%, and the compressive strength and flexural strength of pure PMMA were improved by 53.5% and 56.7%, respectively. When the concentration of the nanocomposite in the PMMA resin was 1.4 wt%, the antibacterial rate was 92.1%. Overall, synergistically reinforcing PMMA composite resin with a multi-dimensional nanocomposite structure provided a new perspective for expanding not only the application of resins in clinical settings but also the research and development of new composite resins.

## 1. Introduction

Polymethyl methacrylate (PMMA) has been widely used as a thermoplastic material in oral medicine since the 1940s for applications involving denture bases, temporary crowns, denture re-lining and restorations, cleft palate obturators, and molds for treatment planning [1]. Although PMMA has many advantages as a biomaterial, such as being inexpensive, light weight and bio-compatible, its low impact resistance and performance defects (e.g., easily fatigues) greatly limit its widespread application [2]. As a denture base, it can easily be coated by plaque, such as Streptococcus mutans and Candida albicans [3] in the complex microbial environment of the oral cavity to form a plaque biofilm. Thus can lead to dental caries, denture stomatitis and other oral mucosal diseases [4]: these performance limitations prevent PMMA from being an ideal denture base.

A variety of nanomaterials, including nanoparticles, nanosheets, and nanotubes, have been incorporated into PMMA to improve its physical and chemical properties [5]. However, the ability of nanomaterials to improve these properties depends on the material type, size, content, surface charge, and matrix–matrix reaction [6]. Zbigniew Raszewski et al. [7] added nano-silica (SiO_2_) to thermally cured PMMA and found that the surface hardness and elastic modulus of the PMMA were improved compared to PMMA alone, with this effect strongly dependent on the physical orientation of the filler particles and the capacity to adhere to the resin matrix. However, the flexural strength of PMMA was negatively affected due to the appearance of agglomeration. In addition, Reem Abualsaud [8] et al. showed that a strong bond was formed between the SiO_2_ and PMMA after salinization, which enhanced the interfacial shear strength between the nanomaterial and the matrix and increased the flexural strength of the composite PMMA compared to the untreated PMMA. Mohanmmed M Gad [9] et al. used zirconia nanoparticles (ZrO_2_NPs) treated with a silane coupling agent as the base material and found that the flexural strength of PMMA was enhanced upon the addition of the nanomaterials. However, when the concentration of ZrO_2_NPs increased to 5% and 7.5%, agglomeration of the nanoparticles occurred, resulting in a significant decrease in the compressive strength of the PMMA base material.

The poor antibacterial properties of PMMA significantly limit its application as a prosthetic material in the oral cavity. Bacterial biofilm formation in the oral cavity often leads to the failure of the prosthetic over time [10]. In addition, the growth of bacteria on PMMA can lead to oral infections that can progress into systemic diseases [11,12]. Some studies have shown that more than 60% of patients with Candida-associated stomatitis are due to denture restoration [13]. To overcome these shortcomings, antibacterial drugs and materials have been introduced into denture base materials. Rong Song et al. [14] reported that the antibacterial activity of PMMA was positively correlated with bacterial growth inhibition. Rinaldi et al. [15] modified PMMA with citrate-capped silver nanoparticles, and when the concentration of composite nanoparticles was 3.5 wt%, the area of Candida biofilm decreased from 90% to 6%. Silver nanoparticles are known to have broad-spectrum antibacterial activity. Even at low concentrations, silver nanoparticles can inhibit the growth of Streptococcus mutans, Staphylococcus aureus, Candida, and other microorganisms by selectively destroying cell membranes, producing reactive oxygen species that ultimately destroying DNA structure. Although significant progress has been made in mechanical properties and antibacterial properties in the past, improving the mechanical and antibacterial properties of ideal denture base resins has significant implications in oral medicine applications and beyond.

Hexagonal boron nitride (h-BN) is a two-dimensional nanomaterial that has a similar structure to graphene and has excellent mechanical, chemical, and physical properties. Macroscopically, h-BN is a white powder compared with black graphene which is conducive to its application in stomatology [16,17,18]. BNNPs (boron nitride nanoplatelets), a newly developed nanomaterial, was first investigated in some studies for use as a reinforcement material for dental applications, for which its white color and biocompatibility make it suitable [19]. One study reported on the mechanical properties regarding self-cured acrylic polymethyl methacrylate reinforced with hexagonal boron nitride and stabilized zirconia nanopowders [20]. At the same time, silver nanoparticles (AgNPs) are widely used in many biomedical fields because of their broad-spectrum antibacterial properties, excellent thermal stability and high biocompatibility [21,22,23].

In this work, AgNPs were self-assembled on h-BN nanosheets (h-BNNs) to prepare a two-dimensional zero-dimensional nanocomposite system comprising h-BN nanosheets and AgNPs (h-BN/AgNPs). Then, a series of h-BN/AgNPs PMMA nanocomposites (hAP) with varying concentrations of h-BNNs/AgNPs were prepared after incorporating them into PMMA. Based on the systematic evaluation of the basic structure of hAP, the mechanical properties and antibacterial properties of hAP were further studied. We expect that the mechanical properties and antibacterial properties of PMMA can be synergistically improved through the addition of the composite. We will further elucidate the interaction mechanism between the composite and the matrix.

## 2. Materials and Methods

The h-BNNs/AgNPs nanocomposites were prepared by in situ composite methods. 0.2 g of h-BN (Sigma Chemical Co., St. Louis, MO, USA) powder was dispersed in 100 mL of 50% ethanol solution (Damao Chemical Reagent Factory, Tianjin, China) and ultrasonicated for 10 h to obtain a large amount of exfoliated h-BN. The reduction reaction was carried out with AgNO_3_ solution under the action of the reducing agent N, N-dimethylformamide (DMF) (Fuyu Fine Chemical Co. Ltd., Tianjin, China), and the ultraviolet light with λ = 185 nm was assisted irradiation for 15 min to accelerate the growth of metal nanoparticles. The grown silver nanoparticles are embedded in the layered h-BN nanosheets (Figure 1) to form h-BNNs/AgNPs composites, and the color of the solution changes from milky white to light reddish-brown.

The composites were vacuum dried after standing for 2 h and then subjected to characterization testing. The structure and phase composition of the samples were characterized by an X-ray diffractometer (XRD, Smart APEX II, BRUKER, Bremen, German). The Raman Spectra Determination was performed with a Raman microscope (LabRAM HR Evolution, HORIBA Jobin Yvon S.A.S, Paris, France) under an Olympus microscope, with a 532 nm (6 mV) laser, which obtained by scanning 32 times in two seconds. Absorption spectra of h-BNNs dispersions and h-BNNs/AgNPs were recorded at 200–800 nm with a UV spectrophotometer (Specord 50PLUS, Analytik, Jena, Germany). AFM samples were obtained by pipetting an aqueous suspension of sonicated h-BNNs/AgNPs samples onto a smooth mica carrier sheet. The suspension was evenly spin-coated on the mica substrate at 6000 rpm. Atomic force microscopy images (AFM) were acquired using an atomic force microscope (Multimode8, BRUKER, Billerica, MA, USA). For AFM analysis, a silicon tip on a nitride rod was used with ScanAsyst-air contact mode (resonance frequency 50/90 kHz).

The samples were grouped according to the h-BNNs/AgNPs nanocomposites with different mass fractions, which were hAP containing 0, 0.2, 0.6, 1.0, 1.4, 1.8 wt% h-BNNs/AgNPs in sequence. The compressive strength test (CS) of PMMA is carried out in accordance with the International Organization for Standardization ISO 20795-1 standard. The samples were made in cylindrical metal molds (4 mm diameter × 6 mm height). After preparation, they were placed into distilled water at 37 °C for 24 h for a compression test. The samples for compression testing were loaded on a universal testing machine (R Controller, Testresources, Shakopee, MN, USA) with a crosshead speed of 1 mm/min. The fracture surface of the sample was sprayed with gold, and was observed with a scanning electron microscope (SEM, jsm-5600lv, jeol, Tokyo, Japan). 

For flexural strength testing, samples with dimensions (65 mm long × 10 mm wide × 3.3 mm thick) were made according to the international standard ISO 20795-1. Six groups of samples were prepared in total. The samples were stored in deionized water for 24 h, they were subjected to a three-point bending test on a universal testing machine with a transverse head speed of 0.5 mm/min. The samples were placed on rollers at a distance of 15 mm from the center, the entire unit was installed under the universal testing machine, and the stress-applying rods were fixed on the upper part. The center of the sample was made to coincide with the center of the distance between the two rollers. A load was applied with a “T” bar in the center of the specimen until a fracture occurred, at which point the maximum force (F) required to produce a specimen fracture was recorded. 

Cylindrical samples with dimensions (4 mm diameter × 6 mm height) were used and, after drying, stored in test tubes at 37 ± 2 °C, with each group having five samples. After 24 h, samples were removed and weighed to an accuracy of 0.0001 g using an analytical balance. The 24 h drying cycle was repeated until a constant mass (m_1_) was obtained (mass change less than ±0.001 mg). Specimens from each group were immersed in test tubes filled with distilled water, wrapped in aluminum foil to avoid light exposure, and placed in an incubator at 37 ± 1 °C. Sample weights were recorded every 24 h until constant weight (m_2_) was reached after 7 d. Every 24 h, the specimens were removed from the solution, wiped with a paper towel to remove excess solution, and returned to the solution immediately after weighing. At the end of the soaking period, the specimens were dried as previously described until the specimens reached a constant mass (m_3_). We then calculated the solubility and water absorption values of the samples. 

The sample surface’s static contact angle (CA) was measured using a contact angle meter (SZ-CAMB, Shanghai, China) at room temperature. During the test, the test liquid was placed on the sample surface using a standard microsyringe, and a camera captured the image. In order to obtain an accurate CA value, the contact angle was determined using the ring method, five different positions on the sample surface were measured, and the average value was taken as the test result.

The bacterial plate counting method evaluated the antibacterial ability of the control and experimental groups against Streptococcus mutans (S. mutans, ATCC^®^700610). Streptococcus mutans were placed in BHI broth and cultured in 5% CO_2_ incubator at 37 °C for 12 h. The bacterial suspension was diluted to 1:100 with fresh BHI for further use. Each group took one sample (10 mm in diameter × 2 mm in height) for cleaning, drying, and sterilization. We dropped 0.1 mL of Streptococcus mutans suspension at a concentration of 106 CFU/mL on the surface of the plate specimen, and covered with a polyethylene film (30 × 30 mm), so the suspension forms a thin and uniform liquid film. Specimens were incubated at 90% relative humidity at 37 °C for 24 h. The specimen and polyethylene membrane surface was then rinsed with 20 mL PBS. We diluted 0.1 mL of washing solution 100 times with PBS, spread 100 μL of the diluted washing solution evenly on the agar plate, incubated at 37 °C for 24 h, and counted and analyzed the colonies on the plate. The original bacterial solution that did not grow on the surface of PMMA was treated as a blank control group according to the above steps. Measurements were repeated three times for each group. Next, the sample preparation method is the same as the above method: after incubating the bacterial solution and the sample together for 12 h in the incubator, Streptococcus mutans adhered and aggregated on the surface of PMMA. Non-adherent bacteria on the sample surface were washed with PBS. 

The bacterial sample was fixed with Gluta fixative, and the sample’s surface was sprayed with gold to obtain a scanning electron microscope image of the sample surface. The Nitrotetrazolium Blue chloride (NBT) reduction method measured the intracellular oxidative stress. The bacterial suspension and the samples with the addition of 0 wt% and 1.4 wt% of the complex, respectively, were placed in an incubator at 37 °C for 7 d. Taking 0.1 mL of bacterial solution and add 0.5 mL of NBT, react at 37 °C for 30 min, we added 0.1 mL of hydrochloric acid with a concentration of 1 mol/L to stop the reaction, centrifuged the reacted suspension for 10 min, removed the supernatant, and obtained the precipitate. The cell bodies were treated with dimethyl sulfoxide (DMSO) to extract the reduced NBT from the cells, diluted with PBS. Finally, the absorbance was measured at 575 nm using a UV spectrophotometer.

The in vitro cytotoxicity of 6 groups of PMMA was evaluated by detecting the viability and morphology of fibroblasts (L-_929_, Shanghai Cell Bank, Chinese Academy of Sciences, Beijing, China). The sample preparation method was the same as before, with 3 replicate groups for each group, a total of 18 samples (10 mm in diameter × 2 mm in height). Samples were stored in deionized water for 24 h and thoroughly sterilized before detection. According to ISO standards, the extraction medium was prepared with DMEM serum-free cell culture medium with a specific surface area/volume ratio of 3 cm^2^/mL, and the eluate was prepared by incubating at 37 °C and 5% CO_2_ for 3 d. The cells were seeded in a 96-well cell culture plate with a cell culture medium concentration of 2 × 10^5^ cells/mL well and adhered to the cells after incubation for 24 h. 100 μL of all PMMA extracts and 10 μL of CCK-8; Sigma-Aldrich, Yeasen Biotech Co., Ltd., Shanghai, China) were added to the plate per well. They were incubated for 24, 48, and 72 h in a humidified incubator at 37 °C, 5% CO_2_, and we measured the absorbance at 490 nm by enzyme labeling. The experiment for each group was independently replicated 3 times.

## 3. Results

### 3.1. Characterization

The effective combination of h-BNNs and AgNPs is the fundamental premise for realizing a multidimensional nanocomposite system to enhance the mechanical and antibacterial properties of PMMA. The h-BNNs/AgNPs nanocomposites were characterized by X-ray diffraction (XRD), the spectra of which are shown in Figure 2A. After comparing with the standard card (PDF#96-201-6171), prominent h-BN diffraction peaks were observed at 2θ = 26.9°, 41.82°, 44.16°, 51.02°, 55.34°, and 76.32°. In addition, the XRD spectra of the h-BNNs/AgNPs composites featured a characteristic peak of Ag (111) at 2θ = 38.32°, which corroborated the successful combination of the AgNPs with the h-BNNs. The nanocomposites were also characterized by Raman spectroscopy, and the Raman spectra are shown in Figure 2B. Ultrasound increased the interlayer spacing of the h-BNNs, and there was a characteristic Raman peak of E2g (B-N) vibration observed at 1378 cm^–1^; no other element-related Raman peaks were observed. The addition of AgNPs to the h-BNNs resulted in a red-shift of the h-BN Raman peak to 1582 cm^–1^. This likely occurred because the Ag atoms occupied random sites in the h-BN lattice, causing the van der Waals forces and band gap between the h-BNNs and Ag in the nanocomposite system to change. In addition, the interface reactions occurred between B-N bonds and AgNPs at the interface, demonstrating the successful combination of the h-BNNs and AgNPs in the nanocomposite system. In the UV-Vis spectrum of the nanocomposites, the central absorption peak of the h-BN appeared around 236 nm, which corroborated the formation of h-BNNs (Figure 2C). To further study the structure of the hybrid nanocomposite system, we used atomic force microscopy (AFM) to analyze the three-dimensional morphology of the h-BNNs/AgNPs nanocomposite. From the AFM images (Figure 3), the thickness of the h-BNNs/AgNPs sheet structure was approximately 4 nm. In addition, AgNPs grow in situ at the edge of h-BNNS due to the adsorption of electrostatic force. The fracture surface morphology of the 0 wt% hAP and 1 wt% hAP was compared by scanning electron microscopy (SEM). Compared to the unmodified PMMA, the fracture surface of the 1 wt% hAP had fewer cracks. However, the h-BNNs/AgNPs nanocomposite exhibited a certain agglomeration phenomenon on the fracture surface.

### 3.2. Mechanical Properties and Surface Properties

The compressive strength of each sample was calculated according to the following equation:(1)$$C=\frac{4p}{{\pi d}^{2}}$$
wherein C is the compressive strength (MPa), p is the maximum applied load (N) measured, and d is the diameter of the sample (mm). The compressive strength of each nanocomposite is shown in Figure 4A. Upon the addition of the h-BNNs/AgNPs to the PMMA, the compressive strength of the resulting nanocomposites was higher than that of the unmodified PMMA, with the compressive strength of the PMMA containing 1% h-BNNs/AgNPs being the highest. (Figure 4A), When the composite content was 0.2 wt%, 0.6 wt%, 1.0 wt%, 1.4 wt%, and 1.8 wt%, the compressive strength increased by 8.5%, 26.5%, 53.5%, 16.3%, and 6.5%, respectively.

The flexural strength (σ, MPa) of each nanocomposite sample was calculated according to the following formula:(2)$$\sigma =\frac{3\mathrm{FL}}{2{\mathrm{BH}}^{2}}\;$$
wherein F is the maximum load (N) applied to the center of the sample to make it break, L is the distance (mm) between the two supports under the surface tension, B is the width (mm), and H is the thickness (mm) between the two surfaces of the test piece. Figure 4B compares the average flexural strength (MPa) of the PMMA resin in each test group. The PMMA composite resin containing 1% h-BNNs/AgNPs had the highest average flexural strength, which was 56.7% higher than the unmodified PMMA. The PMMA resin composite containing 0.6 wt% h-BNNs/AgNPs had the second-highest flexural strength; it increased by 26.7% more than the 0 wt% h-BNNs/AgNPs.

Next, the water absorption and dissolution capacities (μg/mm^3^) of each sample were calculated using the following Formulas (3) and (4), where the volume was calculated using the equation V = π(d/2)2h.
(3)$$\mathrm{Wsp}=\frac{m_{2}-m_{3}}{V}$$
(4)$$\mathrm{Wsl}=\frac{m_{1}-m_{3}}{V}$$

In both equations, W_sp_ is the water absorption value, W_sl_ is the dissolution value, and V is the volume. Figure 4C,D show each sample’s average solubility and water absorption capacity measured on the days 1, 7, 14, and 28. As shown in Figure 4C,D, there was no significant difference in the water solubilities and sorption between the hAP nanocomposites on the first day of the experiments. The solubility increased with both prolonged soaking time and increasing concentration of h-BNNs/AgNPs. The solubility of 1 wt% h-BNNs/AgNPs was lowest on day 28 and increased as the concentration of h-BNNs/AgNPs increased. The water absorption capacity of each sample increased over time throughout the experiment, and the nanocomposites having an overall lower water absorption capacity than the control (unmodified PMMA). However, all values met the International Standards Organization (ISO) specification number 20795–1 requirement.

As shown in Figure 5, the contact angle of the nanocomposites gradually increased as the concentration of the h-BNNs/AgNPs increased, but the measured values of each sample were all less than 90°. Among them, the hAP containing 1 wt% h-BNNs/AgNPs had the highest contact angle (86.85°). However, over time, the contact angle of each group decreased, indicating a certain degree of hydrophilicity, When the test lasted for 10 min, the contact angles of each group were 31.2°, 38.1°, 54.2°, 67.6°, 60.4°, and 59.8°.

### 3.3. Biological Evaluation

The antibacterial rate (AR) of each resin sample was calculated according to Equation (5) below:(5)$$\mathrm{AR}=\frac{N_{0}-\mathrm{Nx}}{N_{0}}\times 100\%$$
where in N_0_ is the number of colonies corresponding to the blank control group, and Nx is the number of colonies corresponding to the nanocomposites (x = 0 wt%, 0.2 wt%, 0.6 wt%, 1.0 wt%, 1.4 wt%, 1.8 wt%). Figure 6 and Figure 7A show the effect of the nanocomposite (including h-BNNs/AgNPs concentration) on the loss of Streptococcus mutans viability. While the experimental group showed a more substantial antibacterial effect. As the concentration of the h-BNNs/AgNPs in the nanocomposites increased, the antibacterial activity increased, The 0 wt%, 0.2 wt%, 0.6 wt%, 1.0 wt%, 1.4 wt%, and 1.8 wt% h-BNNs/AgNPs nanocomposites are shown in Figure 7A. The antibacterial rates of PMMA of were as follows: 18% (Figure 6A), 57.1% (Figure 6B), 64.5% (Figure 6C), 83.2% (Figure 6D), 92.1% (Figure 6E), and 85.2% (Figure 6F).

The effect of the PMMA resins on the viability of L_929_ cells was evaluated using the CCK-8 assay (Figure 7C). The results show that the viabilities of the L_929_ cells cultured with the leaching solution containing the PMMA with h-BNNs/AgNPs were all greater than 80% at 24,48,72 h. When treated for 72 h, the cell survival rates of PMMA extracts of 0 wt%, 0.2 wt%, 0.6 wt%, 1.0 wt%, 1.4 wt%, 1.8 wt% h-BNNs/AgNPs nanocomposite system were 98.40%, 97.1%, 95.1%, 96%, 97%, 92.2%, respectively.

Figure 8 displays the images of the co-culture surface of hAP and bacterial solution. Specifically, Figure 8A (4000× magnification) and Figure 8B (8000× magnification) show the images of the 0 wt% hAP, while Figure 8C (8000× magnification) and Figure 8D (15,000× magnification) shows the images of the 1.4 wt% hAP. Compared to the 0 wt% hAP, incubated the bacteria with the 1.4 wt% hAP led to significantly lower numbers of bacteria, and the bacterial morphology changed. In the 8000-times scanning electron microscope image, the morphology of Streptococcus mutans in the control group was clear, and the surface of the bacteria was smooth and continuous without defects (Figure 8A,B). Although the shape of the bacteria in the experimental group was clear, most of the cells had rough surfaces with bumps (Figure 8C,D).

AgNPs can induce the production of reactive oxygen species (ROS), which can induce intracellular oxidative stress and cause damage to the bacterial structure. To assess the ability of the nanocomposites to produce ROS in cells, the nitro blue tetrazolium reduction (NBT) method [24] was used to measure oxidative stress in bacterial cells in response to incubation of the cells with the nanocomposites, and the results are shown in Figure 8B. In the control group, the amount of intracellular ROS was negligible; however, the intracellular pressure of the cells incubated with the nanocomposites was higher compared to the control group. Therefore, it was speculated that there was a significant correlation between the generation of ROS and the antibacterial activity of hAP.

## 4. Discussion

In this work, we sought to improve the mechanical properties and antibacterial properties of PMMA by introducing a hybrid system comprising h-BNNs and AgNPs into resin, and the nanocomposite resins were characterized by XRD, Raman and UV-vis spectroscopy, AFM, and other techniques. These methods confirmed the successful synthesis of the h-BNNs/AgNPs system and the addition of the composite to the denture base material [25].

It is well-known that h-BN and graphite are isostructural [26], and the exfoliated h-BNNs has a higher specific surface area and bond better with the matrix material. We observed that, as the concentration of the h-BNNs/AgNPs system reached 1 wt% amount, the flexural strength and compressive strength of the PMMA resin was significantly improved over the unmodified PMMA. The compressive strength of the acrylic resin containing 1 wt% h-BNNs/AgNPs was 53.5% higher than that of the unmodified PMMA, while the flexural strength increased by 56.7% over the control. The improved mechanical properties might be related to the good bonding strength between the nanocomposite and matri and uniform distribution of h-BNNs/AgNPs nanocomposite particles throughout the PMMA resin. The nanoparticles were capable of filling in voids surrounding the polymer chains, and when the resins were subjected to external stress, the load could be transferred from the matrix material to the nanofiller. This prevented the propagation of cracks due to the strong interfacial interactions between the nanofillers and the resin matrix. However, when the concentration of the nanocomposite within the polymer resin was high, the mechanical properties decreased due to the agglomeration of the nanomaterials, leading to discontinuities in the resin matrix. In the presence of multiple agglomerates, the stress is concentrated at each agglomeration site, resulting in reduced mechanical properties.

While improving the mechanical properties of the denture base acrylic resin, it is important that the composites used to modify the resin do not deleteriously affect the physical properties of the resin, such as water absorption and solubility. Acrylic resin dentures have a tendency to absorb water, resulting in dimensional denture changes and polymer hydrolysis. The water absorption capacity of a denture base polymer should be less than or equal to 32 μg/mm^3^, as recommended by ISO156715. In our studies, the water absorption capacities of the hAP samples were all smaller than the unmodified PMMA, which was attributed to the ability of the nanomaterials to fill in the voids within the polymer resin, endowing the polymer matrix with a more compact and firm structure with lower porosity. The solubility reflects the mass of soluble substances in the polymer. In denture-based acrylic resins, these soluble substances include initiators, activators, plasticizers, and residual monomers [27]. Previous studies have shown that the water absorption capacity of different types of acrylic resins can reach 10–25 μg/mm^3^. In this work, the solubility of the nanocomposite PMMA resins, on average, met the requirements of ISO156715. The results also showed that the water absorption and solubility values were directly related to the concentration of the nanoparticles, as the solubility and water absorption capacity increased as the concentration of the nanoparticles increased. This adverse effect resulted from the aggregation of the nanoparticles at high concentration. Overall, the modified PMMA exhibited a more hydrophobic behavior than the unmodified PMMA.

It has been demonstrated that AgNPs can interact with bacterial cells in different ways [28,29]. AgNPs attach to the bacterial cell membrane by bonding to sulfhydryl groups within membrane proteins, causing disruption to the integrity of the membrane. After penetrating the cell membrane, AgNPs damage intracellular DNA and induce the production of ROS, causing oxidative stress responses. In this study, the h-BNNs/AgNPs composites not only significantly improved the mechanical properties of PMMA but also endowed the PMMA with an antibacterial effect. For example, the hAP containing 1.4 wt.% h-BNNs/AgNPs had an AR that was 92.1% higher than pure PMMA in the denture base. The uniformly distribution of the AgNPs throughout the PMMA matrix will allow the PMMA resin to exhibit a sustained antibacterial effect with wear during clinical use. This method of filler addition is ideal for routes over coating modification.

## 5. Conclusions

In this paper, we successfully prepared h-BNNs/AgNPs two-dimensional nanocomposites and introduced them into the PMMA resin. The flexural strength and compressive strength of the resulting hAP nanocomposites increased by 56.7% and 53%, respectively, compared to pure PMMA. The nanocomposites did not disturb the water absorption and dissolution capacity of the PMMA, and the hAP nanocomposites demonstrated attractive antibacterial properties. Cytotoxicity tests also corroborated the safety to the human body. The denture base material modified by this dual nanosystem with good antibacterial properties and mechanical strength is an ideal substitute for potential existing common base materials, and also provides a new path for the preparation of more multifunctional dental materials. One of the future research directions is to analyze the performance of the modified denture base under the simulated oral masticatory performance. In addition, for further clinical application, the in vivo properties of the modified PMMA resins need to be further studied.

## Figures and Tables

**Figure 1 biomimetics-07-00138-f001:**
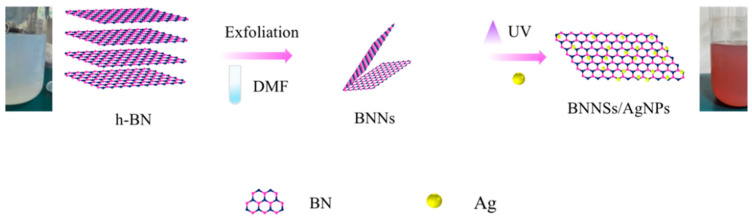
Boron nitride/silver nanocomposite preparation flow chart.

**Figure 2 biomimetics-07-00138-f002:**
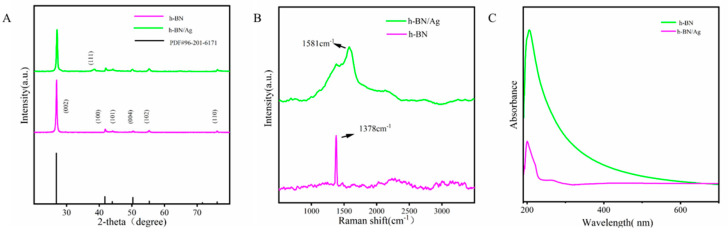
(**A**) XRD patterns of h-BN and h-BNNs/AgNPs (**B**) Raman spectra of h-BN and h-BNNs/AgNPs (**C**) UV spectra of h-BN and h-BNNs/AgNPs.

**Figure 3 biomimetics-07-00138-f003:**
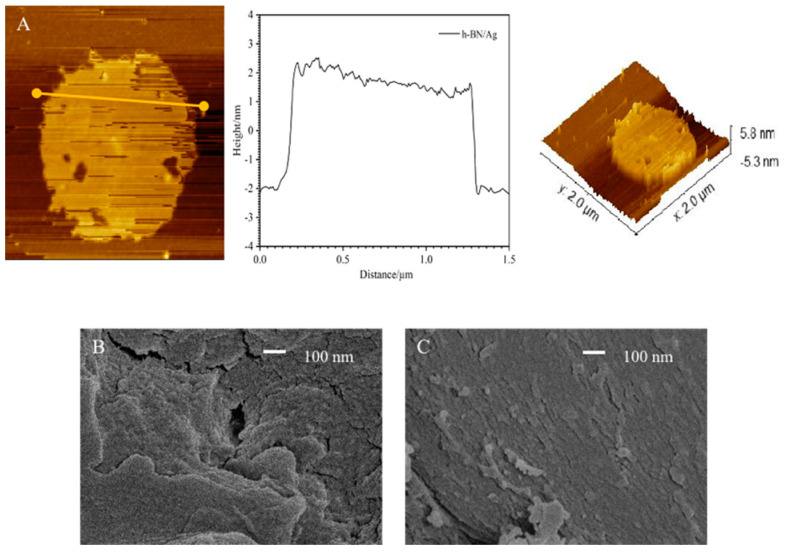
(**A**) AFM image of h-BNNs/AgNPs (**B**) SEM image of hAP (0 wt%) (**C**) SEM image of fracture surface of hAP (1 wt%).

**Figure 4 biomimetics-07-00138-f004:**
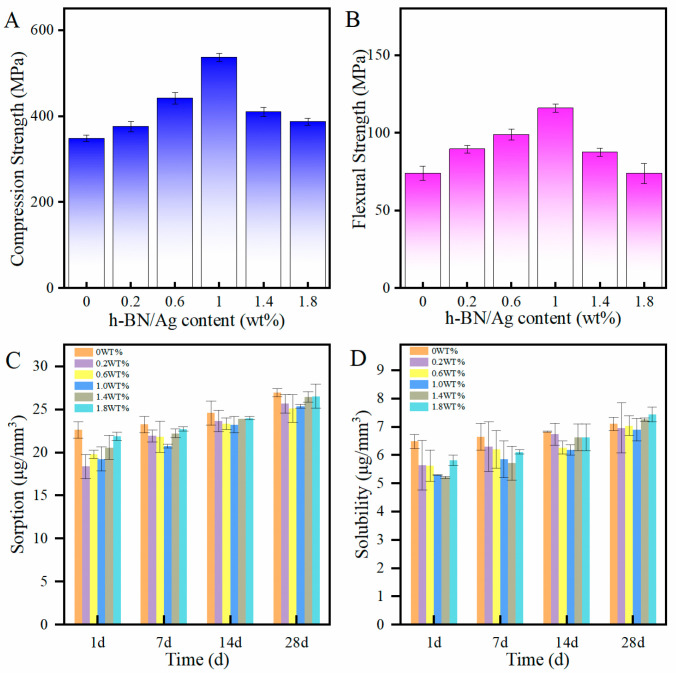
(**A**) Compressive strength of PMMA; (**B**) Bending strength of PMMA; (**C**) Water absorption value of PMMA; (**D**) The dissolution value of PMMA.

**Figure 5 biomimetics-07-00138-f005:**
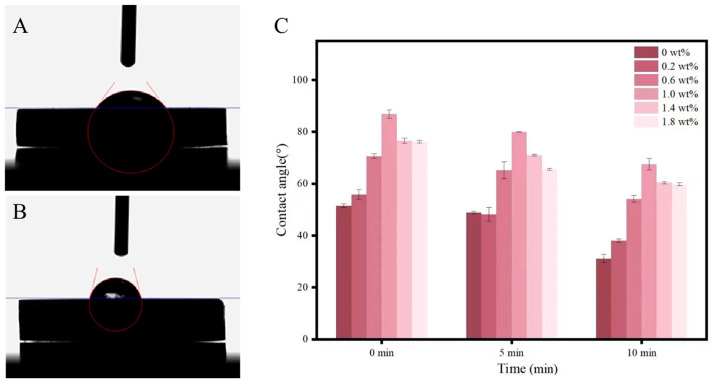
(**A**) Water contact angles of hAP (0 wt%) surfaces; (**B**) Water contact angles of hAP (1 wt%) surfaces; (**C**) Contact angle measurement results of hAP (0, 0.2, 0.6, 1.0, 1.4, 1.8 wt%).

**Figure 6 biomimetics-07-00138-f006:**
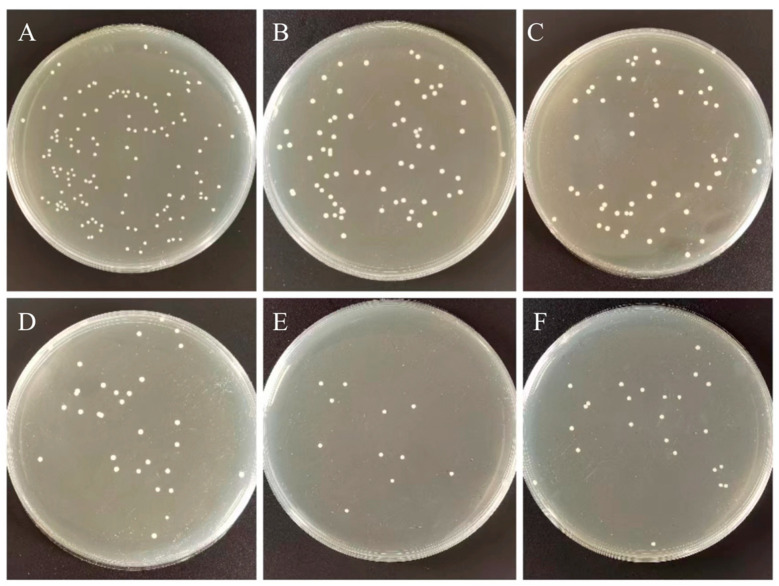
Diagram schematic of plate count method. (**A**–**F**) indicate hAP (0 wt%, 0.2 wt%, 0.6 wt%, 1.0 wt%, 1.4 wt%, and 1.8 wt%).

**Figure 7 biomimetics-07-00138-f007:**
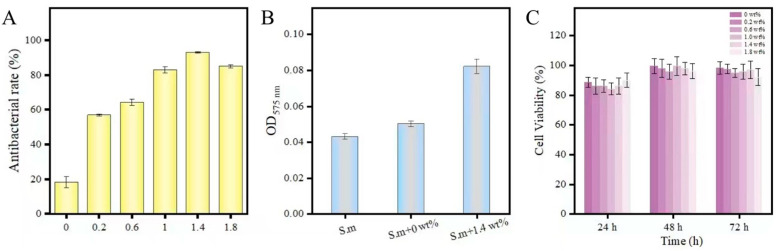
(**A**) Antibacterial rate of hAP (0, 0.2, 0.6, 1.0, 1.4, 1.8 wt%); (**B**) The original bacterial solution, the absorbance of hAP (0 wt%) and hAP (1.4 wt%) at 575 nm; (**C**) Cell viability of L_929_ cells co-cultured with hAP (0, 0.2, 0.6, 1.0, 1.4, 1.8 wt%).

**Figure 8 biomimetics-07-00138-f008:**
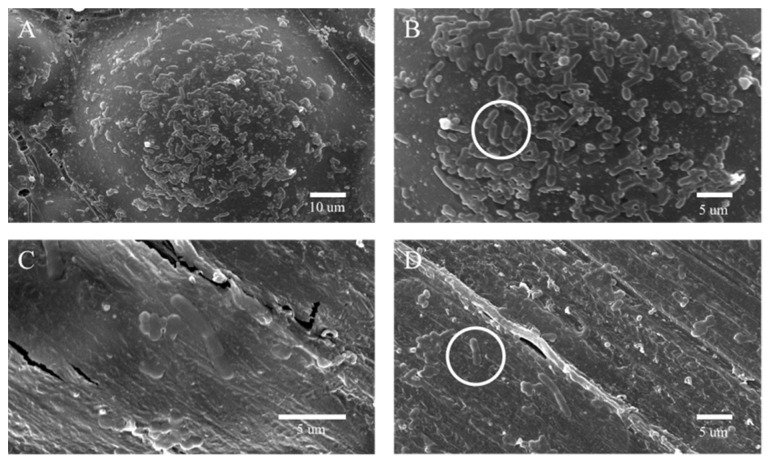
SEM morphologies of S. mutans on the samples after being cultured for 12 h. (**A**,**B**) The co-culture surface of hAP (0 wt%) and bacterial solution. Circle notes in the (**B**) indicate the typical morphology of bacteria before antibacterial test. (**C**,**D**) The co-culture surface of hAP (1.4 wt%) and bacterial solution. Circle notes indicate the change of bacterial morphology after antibacterial test in (**D**).

## Data Availability

All data needed to review the study are present in this paper.

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
