# Peer review of "The Mechanical and Antibacterial Properties of Boron Nitride/Silver Nanocomposite Enhanced Polymethyl Methacrylate Resin for Application in Oral Denture Bases"

_biomimetics, 2022, doi:10.3390/biomimetics7030138_

Round 1
Reviewer 1 Report
Thank you for the submission.
The tile should focus on aim and the conclusions should be alligned with the aim.
There are english errors in the abstract.
Kindly mention the chemical reaction of the nanoparticles and materials incorporated with the PMMA. mention studies on boron material in dental materials and also studies to support the claim that it improves mechanical properties. dvidie methods in sections. sample size calculation and the number of samples used for tests. the sample dimensions.present study groups and tests in a illustration.
The methods should be supported by previous studies. mention the assembly of mechanical testing. provide accurate details of biological testing.
the standardization and testing should be mentioned.
hypothesis is lacking. mention study clinical significance and limitations.
Were the particles coated. Incorporation of nano particles and boron, to the pmma please describe ratios and how long after it was tested. how much to the sample was prepared.
The authors keep mentioning the preparation, however it was not in the study aim.
Revise conclusiosn to be alligned with title and aim.
Author Response
We would like to thank all the reviewers for their time and effort invested in the evaluation of our manuscript. We thoroughly considered all the comments and extensive editorial changes were made in the main text. Please find below or point-by-point replies to the critique accompanied by brief descriptions of changes made in the manuscript
Reviewer 1
Thank you for the submission.
- The title should focus on aim and the conclusions should be alligned with the aim.
Reply 1-1. Thank you for your suggestion. We have revised the title.
- There are english errors in the abstract.
Reply 1-2. Thanks for your suggestion. We have corrected the english errors in the abstract.
- Kindly mention the chemical reaction of the nanoparticles and materials incorporated with the PMMA. mention studies on boron material in dental materials and also studies to support the claim that it improves mechanical properties. dvidie methods in sections. sample size calculation and the number of samples used for tests. the sample dimensions.present study groups and tests in a illustration.
Reply 1-3. Thanks for your reminder. In fact, there is no chemical reaction between the nanocomposite and the PMMA substrate, and their binding relies on electrostatic and van der Waals forces. We have added previous studies of boron nitride nanosheets for enhancement in the background section. The research methodology was presented in three paragraphs, which indeed lacked clarity, and have been revised in the new manuscript. The number of samples as well as the sample parameters are added in the new manuscript. The sample with 0 wt.% nanocomposite addition in the inset is the control group, which is clear.
- The methods should be supported by previous studies. mention the assembly of mechanical testing. provide accurate details of biological testing.
Reply 1-4. Thank you very much for your suggestion. We have supplemented the ISO standard numbering in the new manuscript, and more biological testing details have been added in the manuscript.
- The standardization and testing should be mentioned.
Reply 1-5. Thanks for your suggestion. We added the national standard number in the article.
- hypothesis is lacking. mention study clinical significance and limitations.
Reply 1-6. Thanks! The hypothesis for the study was present in the introduction section of the original manuscript.
We expect that the mechanical properties and antibacterial properties of PMMA can be synergistically improved through the addition of the composite.
At the end of the article, we add the clinical implications and limitations of the findings.
- Were the particles coated. Incorporation of nano particles and boron, to the pmma please describe ratios and how long after it was tested. how much to the sample was prepared.
Reply 1-7. The nanocomposites are uniformly dispersed in PMMA. We further supplemented the sample preparation size and quantity in the experimental process part of the article. And the addition amount of the complex in each group was described in detail.
- The authors keep mentioning the preparation, however it was not in the study aim.
Reply 1-8. Thank you for your suggesstions. We only mention preparation in materials and methods. We think it is necessary.
- Revise conclusiosn to be alligned with title and aim.
Reply 1-9. Thank you very much for your kind reminding. We revised the conclusion part to make it consistent with the title and research purpose.

Reviewer 2 Report
The authors present a study on PMMA resins with inclusion of boron nitride/ silver nanocomposites. Most of the figures should be improved to achieve a better readability of the paper.
- The caption for figure 1 should give the full name of the systems, not only the abbreviations.
- The three graphs in figure 2 are missing the A B C notation. Font sizes are too small.
- The caption of figure 3 does not mention part C. What is the meaning of the white arrows?
- What does "each group" mean in the caption of figure 4? It should be "MPa" and "µg" instead of "Mpa" and "ug".
- "Contact angle test" as a figure caption does not give any details. The example measurements pictured seem arbitrary.
- The bacterial plate count given in figure 6 is marked with a-f for "each experimental group". How should the reader find out what this is about?
- For the SEM images in figure 8 it is mentioned in line 297 that there is a significant lower number of bacteria with a changed morphology for the 1.4 wt% hAP sample. It should be indicated where this can be seen in the SEM images.
-
Author Response
We would like to thank all the reviewers for their time and effort invested in the evaluation of our manuscript. We thoroughly considered all the comments and extensive editorial changes were made in the main text. Please find below or point-by-point replies to the critique accompanied by brief descriptions of changes made in the manuscript
Reviewer: 2
The authors present a study on PMMA resins with inclusion of boron nitride/ silver nanocomposites. Most of the figures should be improved to achieve a better readability of the paper.
- The caption for figure 1 should give the full name of the systems, not only the abbreviations.
Reply 2-1. We appreciate the suggestion for revision of the caption for figure 1, we already gave the full name of the systems.
- The three graphs in figure 2 are missing the A B C notation. Font sizes are too small.
Reply 2-2. Thank you very much for pointing this out. We modified the font size as well as the title of Figure 2.
- The caption of figure 3 does not mention part C. What is the meaning of the white arrows?
Reply 2-3. Thanks. We modified the caption of figure 3. White arrows indicate cracks, and in order to avoid ambiguity, we deleted the arrows in the figure.
- What does "each group" mean in the caption of figure 4? It should be "MPa" and "µg" instead of "Mpa" and "ug".
Reply 2-4. We apologize for the error of writing. We have corrected these errors.
- "Contact angle test" as a figure caption does not give any details. The example measurements pictured seem arbitrary.
Reply 2-5. Thanks. We have supplemented the relevant tests of contact angle and modified the schematic diagram of contact angle measurement.
- The bacterial plate count given in figure 6 is marked with a-f for "each experimental group". How should the reader find out what this is about?
Reply 2-6. Thanks for your suggestion. We have added the description of figure 6 (a-f) in the article and supplemented the detailed description in Figure 6.
- For the SEM images in figure 8 it is mentioned in line 297 that there is a significant lower number of bacteria with a changed morphology for the 1.4 wt% hAP sample. It should be indicated where this can be seen in the SEM images.
Reply 2-7. Thank you very much for pointing this out. We have made significant white marks for the changes in bacterial morphology in figure 8.And it is further explained in the article.

Round 2
Reviewer 1 Report
the aim of the study is still missing in the abstract. this is a requirement for abstract. you mention what you did. but aim shud be clearly mentioned
english language correction still not apt. "We employed scanning electron microscopy (SEM), X-ray diffraction (XRD), Raman spectroscopy (Raman), Ultraviolet visible spectrum (UV) and Atomic force microscopy (AFM) to investigate the microscopic morphology and composition of PMMA containing different mass fractions of the nanocomposites using". need revision
The keywords should be from the MeSH terms, revise plz
confusing sentence making "Although PMMA has many advantages as a biomaterial because it is inexpensive, light weight, and bio-compatible, its low impact resistance and performance defects (e.g.easy fatigue) greatly limit its widespread application". correct the use of articles and punctuations.
plaque is the proper word not plaques, as it is accumulation of microorganisms
Statements made in the introduction are supported by misplaced references. ref 2 does not support the claim about plaque.
for one example. "As a denture 36 base, PMMA easily attaches to plaques such as Streptococcus mutans and Candida albicans [2] in the complex microbial environment of the oral cavity to form a plaque biofilm, which can lead to dental caries, denture stomatitis, and other oral mucosal diseases 39 [3,4]; these performance limitations limit PMMA from being an ideal denture base" these references do not talk about plaque on PMMA at all. In addition this sentence is too long and tries to articulate too many arguments, but is not effectively doing that.
The poor antibacterial properties of PMMA significantly limit its application as a 59 prosthetic material in the oral cavity. Bacterial biofilm formation in the oral cavity often 60 leads to the failure of the prosthetic over time. In addition, the growth of bacteria on 61 PMMA can lead to oral infections that can progress into systemic diseases. (not supported with evidence)
check all statements in introduction and discussion.
Kindly revise the complete manuscript for english language, sentence correction and correct references for evidence.
Author Response
Thank you for the submission.
1.the aim of the study is still missing in the abstract. this is a requirement for abstract. you mention what you did. but aim shud be clearly mentioned.
Reply 1-1. Thank you for your kindly suggestion. We have added statement on research purpose.
2.english language correction still not apt. "We employed scanning electron microscopy (SEM), X-ray diffraction (XRD), Raman spectroscopy (Raman), Ultraviolet visible spectrum (UV) and Atomic force microscopy (AFM) to investigate the microscopic morphology and composition of PMMA containing different mass fractions of the nanocomposites using". need revision
Reply 1-2. Thanks for your suggestion. We have corrected the english errors in the abstract.
- The keywords should be from the MeSH terms, revise plz
Reply 1-3. Thanks for your reminder.We have modified it, the keywords are from the MeSH terms.
4.confusing sentence making "Although PMMA has many advantages as a biomaterial because it is inexpensive, light weight, and bio-compatible, its low impact resistance and performance defects (e.g.easy fatigue) greatly limit its widespread application". correct the use of articles and punctuations.
Reply 1-4. Thank you very much for your suggestion. We have modified the sentence to make it smooth and corrected the articles and punctuations.
5.plaque is the proper word not plaques, as it is accumulation of microorganisms
Reply 1-5. Thanks for your suggestion. We have revised it.
- Statements made in the introduction are supported by misplaced references. ref 2 does not support the claim about plaque.for one example. "As a denture 36 base, PMMA easily attaches to plaques such as Streptococcus mutans and Candida albicans [2] in the complex microbial environment of the oral cavity to form a plaque biofilm, which can lead to dental caries, denture stomatitis, and other oral mucosal diseases 39 [3,4]; these performance limitations limit PMMA from being an ideal denture base" these references do not talk about plaque on PMMA at all. In addition this sentence is too long and tries to articulate too many arguments, but is not effectively doing that.
Reply 1-6. Thanks! I'm very sorry for this error.References have been corrected and relevant evidence was supplemented.
- The poor antibacterial properties of PMMA significantly limit its application as a 59 prosthetic material in the oral cavity. Bacterial biofilm formation in the oral cavity often 60 leads to the failure of the prosthetic over time. In addition, the growth of bacteria on 61 PMMA can lead to oral infections that can progress into systemic diseases. (not supported with evidence)
Reply 1-7. Thanks for your reminder.Relevant evidence was supplemented in reference.
8.Kindly revise the complete manuscript for english language, sentence correction and correct references for evidence.
Reply 1-8. Thank you very much for your kind reminding. We carefully examined the full text.
